# Long-Term Performance of Blue-Green Roof Systems—Results of a Building-Scale Monitoring Study in Hamburg, Germany

Michael Richter * and Wolfgang Dickhaut

Department of Environmentally Sound Urban and Infrastructure Planning, HafenCity University Hamburg, 20457 Hamburg, Germany
* Correspondence: michael.richter@hcu-hamburg.de

**Abstract:** For the first time, a long-term monitoring study with different full-scale blue-green roof (BGR) types was conducted. Within a pilot project from Hamburg's Rainwater InfraStructure Adaptation (RISA) framework, four different BGR types were built in 2015 for long-term evaluation and comparison with each other. The test site was created to find out to what extent BGRs are able to improve hydrological performance and if increased water supply affects vegetation development and species diversity. Therefore, the roofs were equipped with hydrologic monitoring systems, their retention performance was evaluated, and vegetation analysis was conducted. During 2017–2023, between 64 and 74% of the precipitation was retained on the roofs, and in the summer months there was hardly any outflow from the roofs. For single (heavy) rain events, high retention capacities, low outflow intensities, and high detention times were demonstrated. On the BGRs where rainwater is permanently stored on the roof, the vegetation species' composition changed in the long term, resulting in an increase in biodiversity. The studied BGRs are effective in reducing flood risk from heavy rain events and can increase evaporative cooling and biodiversity. Therefore, such BGRs are a blue-green infrastructure with far-reaching positive effects.

**Keywords:** green roof; blue-green roof; blue-green infrastructure; climate change adaptation; urban water cycle; WSUD; sponge city

## 1. Introduction

Green roofs (GRs) are a specific type of blue-green infrastructure and a well-known element of water-sensitive urban design. Modern types of GRs consist of the same principal elements as can be seen in Figure 1 (without a storage pipe and an outflow control element). The root barrier protects the waterproofing of the roof from penetration by plant roots. For conventional GRs, there is a drainage layer to both retain rainwater and drain away surplus water. The roof slope for GRs is usually between 2% and 5%. Drainage layers can be industrially produced plastic frames as well as gravel layers. Filter layers are often integrated into the system between the substrate and drainage layer to avoid small-sized particulates being washed out. The substrate layer functions as a growing medium for plants by providing a rooting zone and as a water-storage medium. It is typically a lightweight aggregate with both high water-holding capacity due to high porosity and good drainage properties. Substrate depth is an important property that determines water-retention capacity, plant growth, and plant selection. Depending on the substrate layer thickness or the depth to which plant roots can penetrate into the medium, two types of green roofs are distinguished. Extensive GRs have a substrate thickness of 6–15 cm; semi-intensive GRs 15–25 cm; and intensive green roofs have more than 25 cm thick substrate layers. The types of plants used depend on the type of GR and the local climate. On extensive GRs, due to regular drought stress, winter-hard, drought-tolerant, and perennial plants such as sedum species dominate. On intensive GRs, grasses, shrubs and even trees can grow. The positive aspects of GRs have been recognized by intensive research,

especially in recent decades since the 1980s. The reduction in heating and cooling costs and the reduction in the urban heat island (UHI) effect through thermal insulation and shading and cooling through evapotranspiration have been described by [1,2]. GRs reduce and delay stormwater runoff due to (heavy) rain events, which can be attributed to water storage on plant surfaces (interception) and in substrates and evapotranspiration from plants and substrate surfaces [3–5]. Other positive effects of GRs include noise reduction [6], filtering of air pollutants [7,8], and increasing urban biodiversity [9,10] which promotes quality of life and human well-being in urban areas [11]. In the context of climate change adaptation of urban areas, i.e., water-sensitive or sponge cities, the aspects of reducing the UHI effect and reducing the risk of urban flash floods are of particular importance and have been intensively researched worldwide in recent years.

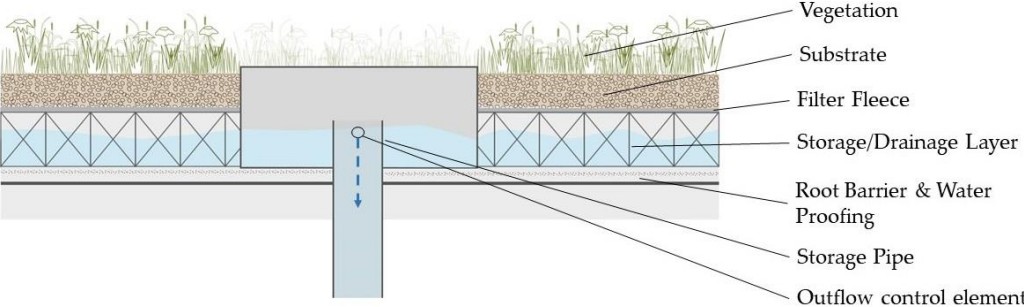

**Figure 1.** Schematic drawing of a blue-green roof.

Cooling of urban areas using GRs is provided by several factors [12]:

- Shading of surfaces or prevention of direct solar radiation;
- (Evapo)transpiration or evaporative cooling by active evaporation from plants (transpiration) and evaporation from water surfaces (evaporation);
- Change in air flow and circulation;
- Insulation effect by substrate and plant layers on roofs;
- Change in albedo of surfaces, thereby increased reflection of solar irradiation compared to darker surfaces;
- Partial absorption of solar irradiance and conversion to biomass (photosynthesis).

Risk reduction of urban flash floods by GRs is mainly provided by the following factors:

- Interception of rainwater on plant surfaces;
- Retention and storage of rainwater in substrates;
- Evaporation of rainwater from plant and substrate surfaces (evaporation);
- Active evaporation of rainwater by plants (transpiration);
- Temporal delay of stormwater runoff through intermediate storage in substrates;
- If necessary, additional rainwater retention through storage, e.g., in cisterns for irrigation.

The hydrological efficiency of GRs depends on the roof slope [13], substrate depth [5], rainfall characteristics such as duration [14] and season [4], soil moisture [15], the age of the roof [16], plant species [17], and the type of growing media [18]. According to the green roof guidelines of the German Research Society for Landscape Development and Landscaping [19], green roofs with a substrate layer thickness of more than 6 cm (extensive green roofs) can retain about 50% of precipitation (runoff coefficient C = 0.5), and intensive green roofs with growing medium thickness greater than 50 cm can retain more than 90% (C < 0.1).

Blue-green roofs (BGRs) have been developed in recent years to increase retention of and delay stormwater runoff and to increase their evapotranspiration performance. Until now, there has been no universal definition of BGRs. The updated edition of the German Green Roof Guidelines [19], which traditionally serves as a model for international guidelines, addresses this technical advancement of GRs (the designation retention roof is used in German for BGRs): "In [...] retention roofs, the water is accumulated and

temporarily stored in the green roof structure, if necessary, also in an additional layer. The runoff is throttled in volume and/or delayed in time under defined conditions". A corresponding schematic drawing of the individual layers is shown in Figure 1. In the case of BGRs, the drainage layer works as a storage layer.

In other publications, BGRs are defined as GRs with an additional layer that stores more water than the (extensive) vegetation requires [20], as a blue roof located beneath a green roof system (and combining their benefits) [21], or, e.g., as green roofs that are explicitly built as part of a stormwater management system [22]. The latter definition would likely include conventional green roofs, as these are often part of drainage design in practice. As a useful distinction in the technical sense, the authors suggest the objective of infiltrating rainwater below the substrate layer and retaining rainwater in a temporary or long-term manner in a storage layer by means of a throttled runoff or gradual release (Figure 1). This means that the level of retention and the intensity of the runoff can be controlled by changing the diameter of the outflow control element and its height above the roof. Thus, the hydrological performance can be set according to the targets. By designing BGRs as slope-free roofs, runoff coefficients for individual rain events are reduced further [23]. So-called "smart blue-green roofs" with prediction-based outlet controls could further increase the hydrological efficiency of BGRs and increase their effects for flood prevention, plant water supply, and evaporation [24]. For these automatically controlled BGRs, the authors will use the term dynamic BGRs as distinct from static BGRs with fixed outlet settings.

The additional stormwater storage in comparison to conventional extensive GRs results in less water discharge in the long term, and in some cases over 90% of the rainfall can remain on the roofs [25]. Depending on the design of the outlet devices, runoff coefficients close to 0 can be achieved even during exceptional and extreme heavy rain events [25,26]. In addition, outflow elements can be designed to control how long the rainwater is stored on the roof for plant water supply or how long it takes for the retention roof to reach its full retention potential again.

Since these types of blue-green infrastructures are still new, and there are as of yet few studies on full-scale BGRs, the aims of the study were the following:

- To create a field test site with different BGR types and find out to what extent they are able to improve hydrological performance, i.e., the potential for flood and heat prevention;
- To determine the extent to which the increased water supply affects vegetation development and species diversity.

## 2. Materials and Methods

The Rainwater InfraStructure Adaptation (RISA) project was developed in cooperation with a local housing company (SAGA), the Hamburg Environmental Agency, Hamburg Wasser (Water Agency), and two green roof companies (Zinco GmbH, Nürtingen, Germany and Optigrün international AG, Krauchenwies-Göggingen, Germany) in the context of the development of a water-sensitive residential area. In addition to various established decentralized urban water management systems such as extensive green roofs, vegetated retention and infiltration swales, infiltration trenches, and retention basins, various BGR types were implemented and built in December 2015.

### 2.1. Study Location

The study site is located in Hamburg-Ohlsdorf (coordinates: 53.61983, 10.01634 (https://www.openstreetmap.org/#map=18/53.61983/10.01634, accessed on 14 June 2023)). Hamburg is the second-largest city in Germany with a population of 1.85 million.

It is located in Northern Germany on the Elbe River (Figure 2) and has an oceanic climate (Köppen-Geiger classification: Cfb), influenced by its proximity to the coast and maritime influences that originate over the Atlantic Ocean.

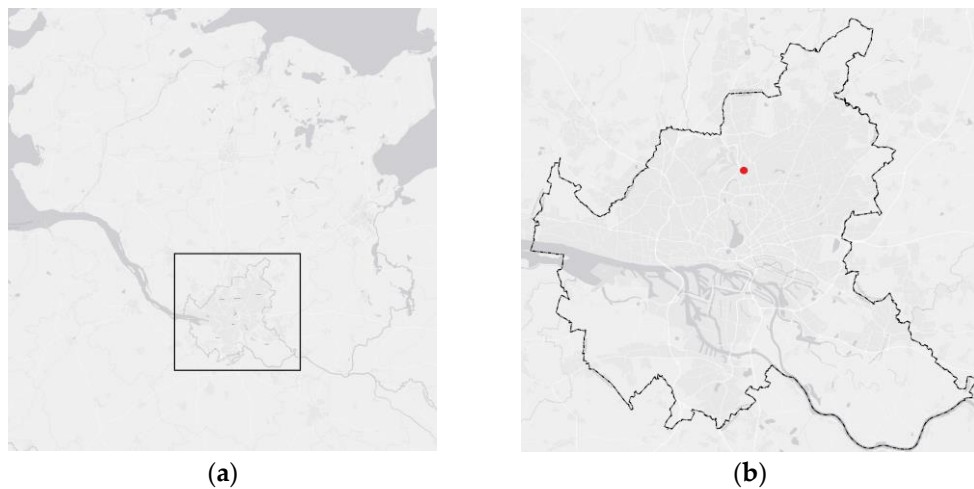

(**a**)            (**b**)

**Figure 2.** Geographic location of Hamburg in Northern Germany (**a**) and location of the study site (red point), a residential area in Hamburg-Ohlsdorf (**b**) (© geographical data provided by Esri, HERE, Garmin, OpenStreetMap contributors, and the GIS user community.

### 2.2. (Blue-)Green Roof Types

On three residential buildings, six different roof types were built in December 2015 and equipped with hydrological monitoring systems for long-term evaluation and comparison with one another. The buildings were three-story apartment blocks made of reinforced concrete construction. On two buildings, four larger areas (220 m$^2$ each, of which approx. 5% were ungreened), and on the other building, two smaller areas (135 m$^2$ each, approx. 8% ungreened) were equipped with different roof types. Different types of BGRs were built on the four larger areas, and an extensive green roof and an ungreened gravel roof were built on the smaller areas for comparative measurement (Figure 3).

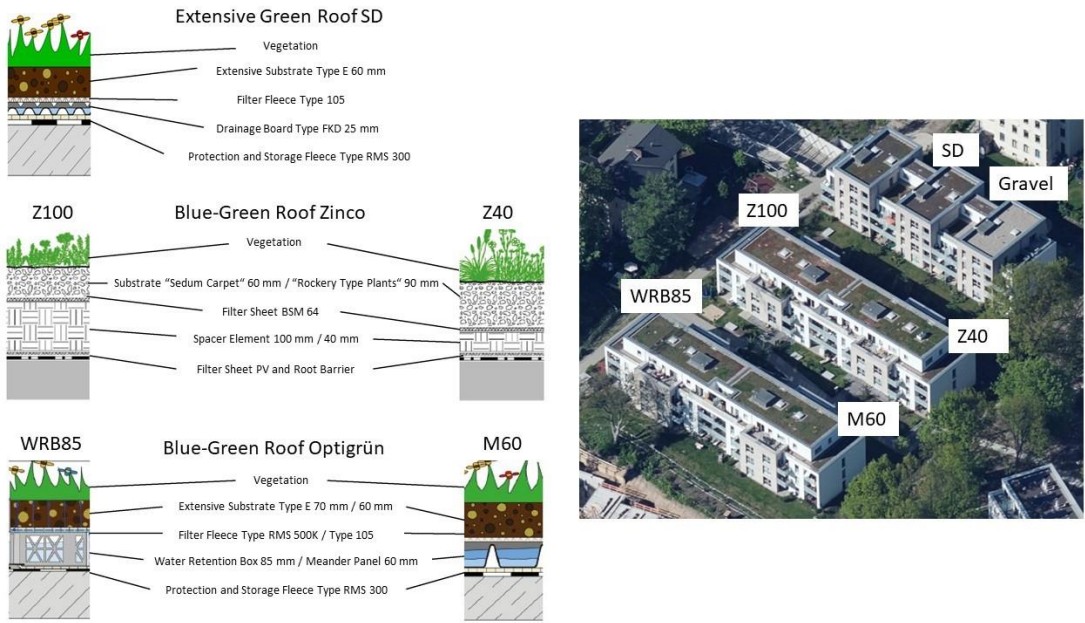

**Figure 3.** Diagrams of the different (B)GR systems, one extensive GR and four types of BGRs on two different buildings. In the 3D view of the residential buildings, the different roof types are marked. (GR system drawings provided by: Optigrün International AG, Krauchenwies-Göggingen, Germany and Zinco GmbH, Nürtingen, Germany; 3D view provided by: Hamburg State Office for Geoinformation and Surveying LGV).

One building is equipped with two different types of Zinco BGRs (Figure 3), with 40 (Z40) and 100 mm (Z100) high storage elements, respectively, and an outflow control element ("runoff limiter RD28") installed at the height of the roof sealing. This results in a throttled runoff, but no long-term water storage under the substrate layer. These roofs have substrate layer thicknesses of 90 mm (Z40) and 60 mm (Z100). They were sown in the spring of 2016 with a typical sedum mixture ("Sedum carpet", Z100) and with a specific regional plant mixture for extensive green roofs with sedums, herbs, and grasses ("Hamburg nature roof", Z40).

On the other building, two different types of Optigrün BGRs were built. The systems "Flow Control Extensive" (WRB85) and "Meander FKM 60" (M60) have storage elements of 85 mm (WRB85) and 60 mm (M60) in height, respectively. On the Meander FKM 60 roof, water flow is further slowed down by the storage layer structure, which has a meandering route from one chamber to another. These two roofs were equipped with a static outflow control at a height of 30 mm above the roof surface and therefore equipped with long-term storage capabilities. These roofs have a substrate layer thicknesses of 80 mm (M60) and 70 mm (WRB85). They were sown in the spring of 2016 with a sedum, herb, and grass mixture.

The four BGR types on these two buildings were designed as 0° roofs to ensure a uniform water level (in case of accumulation).

A gravel roof and an extensive GR, the "Optigrün Economy Roof" (SD) type with a 25 mm drainage element, 60 mm substrate layer, and sedum vegetation, were installed on the third building for comparison purposes. The types are described in more detail in Figure 3 and Table 1.

**Table 1.** Specifications of the different roof types.

| Name | M60 | WRB85 | Z40 | Z100 | SD |
|---|---|---|---|---|---|
| Type | BGR | BGR | BGR | BGR | Extensive GR |
| Substrate height [mm] | 80 | 70 | 90 | 60 | 60 |
| Storage element height [mm] | 60 | 85 | 40 | 100 | 25 (Drainage) |
| Long-term storage [mm] | 30 | 30 | - | - | - |

*2.3. Monitoring, Data Preparation, and Calculations*

2.3.1. Hydrological Monitoring and Data Preparation

The hydrological measurement technology was installed in 2016, and the data for precipitation and runoff used in this study are from July 2017 to March 2023 (with interruptions due to instrument failure, 70 months). Discharge amounts of the BGRs were measured every minute via 1 L steel tipping counters installed in measuring boxes (see Figure 4a, constructed by Umweltgerätetechnik GmbH UGT, Müncheberg, Germany). The boxes that measure runoff from the extensive and gravel roofs record flow data using a dual system with a Thompson weir (water level measurement via KELLER Series 46 X (KELLER Druckmesstechnik AG, Winterthur, Switzerland) submersible capacity transmitter) and 100 mL polycarbonate tipping counters (constructed by UGT, Figure 4b). Rainfall amounts are recorded every minute via a rain gauge based on Hellmann with an integrated 2 mL tipping counter (Lambrecht Precipitation Sensor 15189, LAMBRECHT Meteo GmbH, Göttingen, Germany). The precipitation sensor was placed on a stand at 1.00 m above the gravel roof. In times of precipitation sensor failure, data from the nearby meteorological station from the German Meteorological Service (Fuhlsbüttel, station no. 01975) were used for the calculation of long-term discharge coefficients (monthly, yearly). The monthly data from the Hellmann rain gauge on the roof of the building were corrected for wind-related measurement errors afterwards [27].

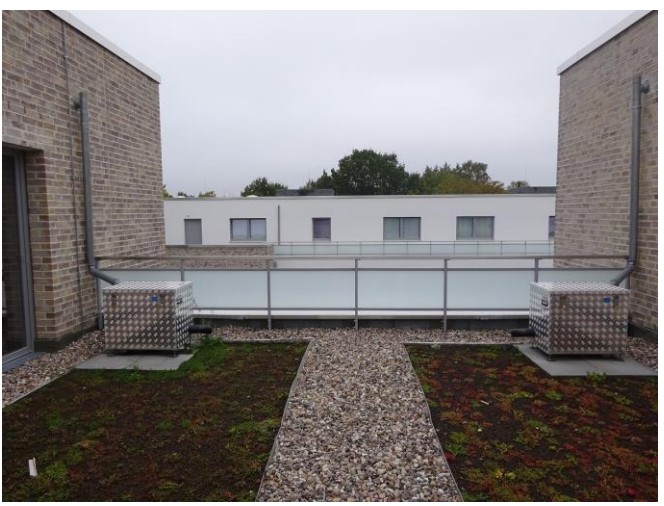
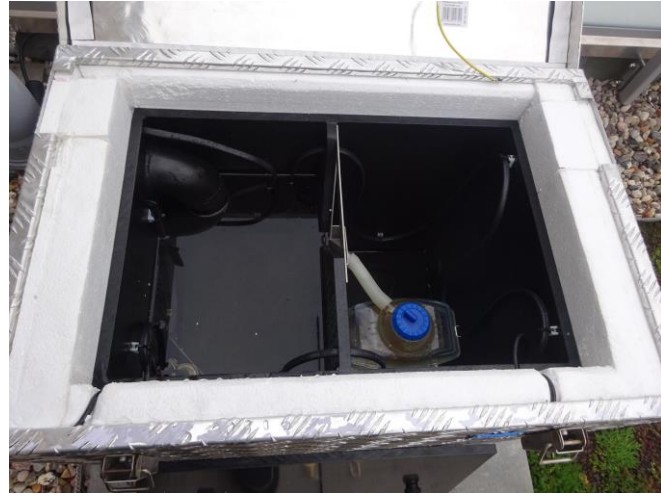

(**a**)                                                                       (**b**)

**Figure 4.** Photographs of the measurement boxes containing the tipping counters (**a**) and of the dual measurement (**b**) system with Thompson weir ((**b**) left side) and polycarbonate counter ((**b**) right side).

In order to generate individual rain events from the data set, it is of great importance to determine the so-called inter-event time in a meaningful way [28]. The inter-event time is defined as the time between two rain events, i.e., the minimum period of time during which no precipitation falls in order to consider two control events as separate from each other. This time span is defined depending on the reference system. It depends significantly on the time required by the respective system for drainage. Rain events were defined as separated from each other in the measured data sets if there was a dry period of at least 6 h without measured precipitation between them. This period was considered a reasonable inter-event time for GR systems in several studies, e.g., [5,13,16,29–31].

In order to calculate discharge amounts for single events, data of the time when precipitation falls plus the discharge during inter-event-time (time of rainfall event + 360 min) were added together. Days on which the daily mean temperature was below 0 °C were not included in the analysis of individual events, since on the one hand, the measurement accuracy of the precipitation amount is reduced in the presence of snow, and on the other hand, it is not possible to say exactly to which precipitation events the runoff can be assigned due to water temporarily freezing on the roof and later melting. Rain events with less than 0.2 mm were also excluded because the measurement accuracy of the system was also in this range. For monthly analysis, months where the runoff amount exceeded the measured rainfall were also excluded. This happened in winter months when snowfall was not measured accurately, but the later discharged melted water was.

### 2.3.2. Performance Evaluation

The retention performance of precipitation on the BGRs was analyzed over different time intervals such as years, months, or individual rain events (minutes to hours). The effect of BGRs relevant for water management practice with regard to design tasks, flood prevention, and the retention of rainfall events was calculated on the basis of runoff coefficients C for the whole measurement duration, for long-term ($C_{lt}$), months ($C_m$), and single events ($C_{se}$). These runoff coefficients were calculated via

$$C = Q/P \tag{1}$$

With Q being the runoff sum [mm] for the time period (month, single event), and P being the associated precipitation sum [mm]. Furthermore, for single rain events, the peak

runoff coefficient $C_p$ was used to evaluate the effectiveness of BGRs to decrease the runoff intensity from the roof:

$$C_p = Q_{max}/P_{max} \qquad (2)$$

Here, $Q_{max}$ is the maximum runoff intensity during the event [mm/min], and $P_{max}$ is the maximum rain intensity [mm/min]. Furthermore, the delay of the start of runoff compared to the start time of the rain [min] and the delay of the peak runoff (max. runoff intensity) in contrast to the maximum rainfall intensity [min] were evaluated. Another effect is the detention time ($T_d$) of the runoff, the time difference between the beginning of a single rain event and the corresponding discharge.

Analysis of variance (ANOVA) was used as a hypothesis test to examine whether mean values of hydrological performance indicators differed from each other. One-way ANOVA was used as a statistical method to test the null hypothesis ($H_0$) that the means of different groups or factors (e.g., season, size of rain event) of runoff coefficients or detention times for each BGR type are equal against the alternative hypothesis ($H_a$) that at least one mean is different from the rest. If there were significant differences, Tukey post-hoc tests were conducted to find out which groups were significantly different from each other. The statistical analyses were carried out with the software IBM SPSS Statistics Version 27.0.0.0.0.

### 2.3.3. Vegetation Analysis

In 2022, vegetation mapping was conducted [32] where all occurring plant species were listed on a homogeneous and representative area, which in this case refers to the whole area of the different (B)GRs (135–220 m$^2$), arranged according to layers (in this case, only the herb layer), and evaluated according to the degree of coverage. Furthermore, the species composition in 2022 was compared with the "theoretical" species composition, meaning the seed mixtures of the different (B)GRs. The analysis was conducted to examine the effects of increased water availability on plant species composition.

## 3. Results

### *3.1. Long-Period Hydrological Performance*

The hydrological performance for longer periods (years/seasons/months) is important for analyzing possible effects on the urban water cycle and potential evapotranspiration values. For the whole measurement period between 2017 and 2023, the runoff coefficients $C_{lt}$ for the different BGR types were 0.26 (M60), 0.32 (WRB85), 0.36 (Z40), and 0.34 (Z100). Since all rainwater which does not drain off the roof will evaporate or be transpired by plants, the long-term evaporation rates for the roof types are 74% (M60), 68% (WRB85), 64% (Z40), and 66% (Z100). These rates are higher than the evaporation from the natural water balance (56%) that could be expected at the site [33].

The monthly average precipitation sums from 2017 to 2023 (Figure 5) indicate seasonal variation with high values in winter (December–February) with 80–106 mm, and relatively dry months from April (35 mm) to June (56 mm). Please note that these values do not reflect the long-term climate.

Monthly average runoff coefficients $C_m$ vary greatly for all different BGR types (Figure 6), and the mean values differ significantly for the BGR types (ANOVA, *p* = 0.002). In general, the wetter and colder months from November to March have mean $C_m > 0.5$ for WRB85, Z40, and Z10. All BGR types have $C_m$ values < 0.3 from April to August. In all months except April (and October for Z100), Z40 and Z100 roofs have higher $C_m$ values than M60 and WRB85 and show similar runoff coefficients in all months except October.

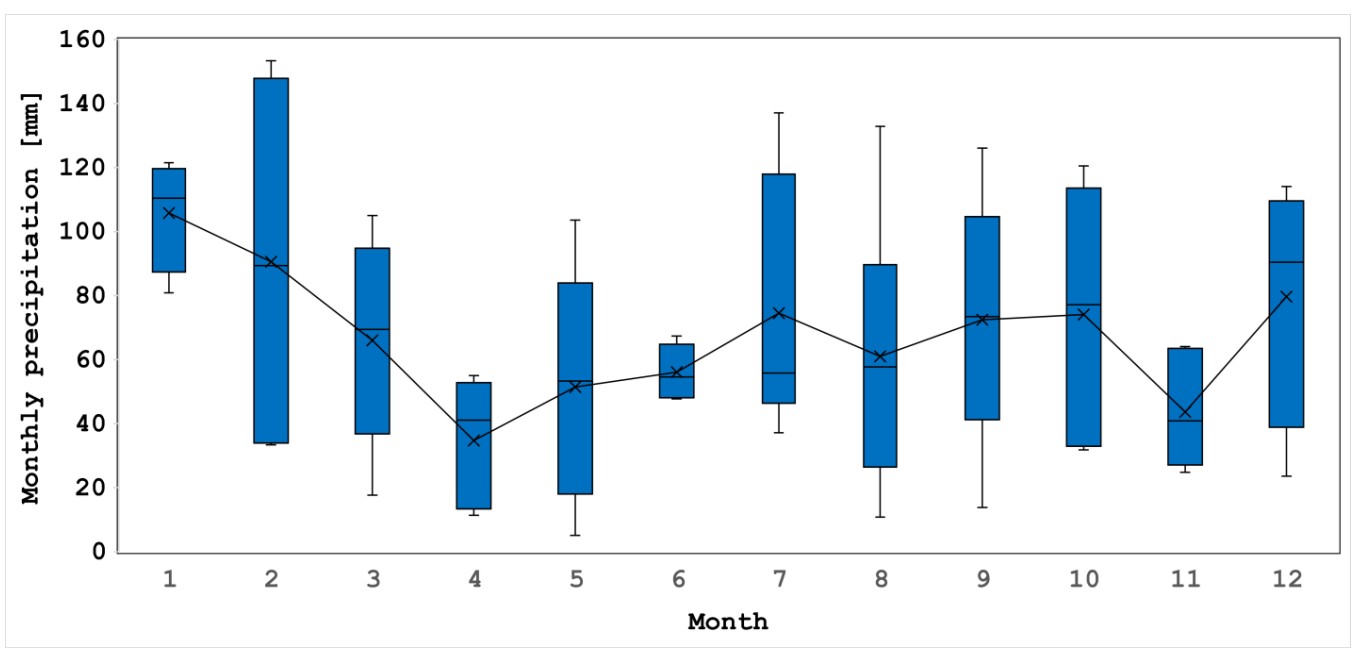

**Figure 5.** Box plots of the monthly average precipitation from 2017 to 2023; mean values are marked by x, and horizontal lines in the boxes are medians.

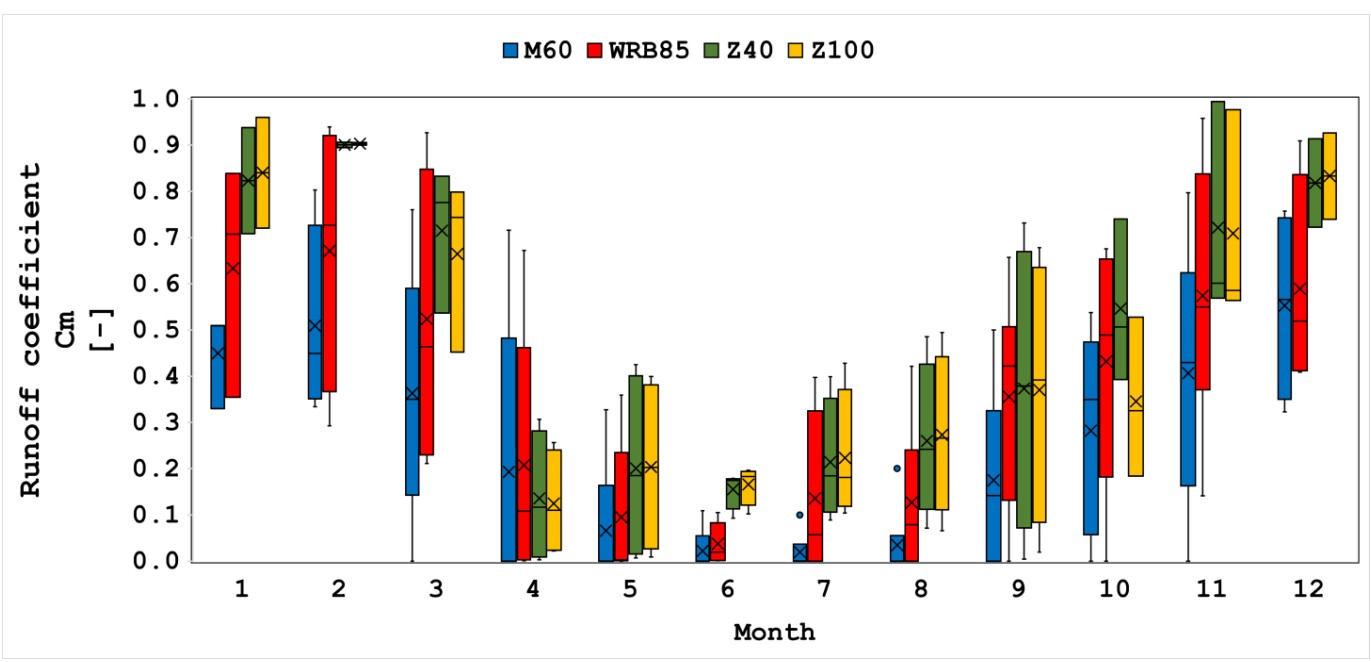

**Figure 6.** Box plots of the monthly average runoff coefficients $C_m$ of the different BGR types from 2017 to 2023; mean values are marked by x, and horizontal lines in the boxes are medians.

The post-hoc Tukey test indicated significant differences in $C_m$ between M60 and Z40 ($p = 0.12$) and Z100 ($p = 0.26$), but there is no significant difference between M60 and WRB85 ($p = 0.176$). In all months except October, M60 was the most effective in rainwater retention. M60 was especially effective in May–August, where mean $C_m$ was lower than 0.1. During the whole measurement period, there were months without any discharge from the different BGRs: 24 (M60) and 7 (WRB85) out of 63 measured months and 2 (Z40) out of 39 months.

### 3.2. Hydrological Performance for Single Rain Events

After data preparation and partial exclusions from the data set, 453 single rain events were analyzed. Of these rain events, about 57% were smaller than 4 mm, and 18% measured from $4 \leq 8$ mm (for details see Figure 7). The highest rainfall amount per event was 56 mm on 24 July 2017.

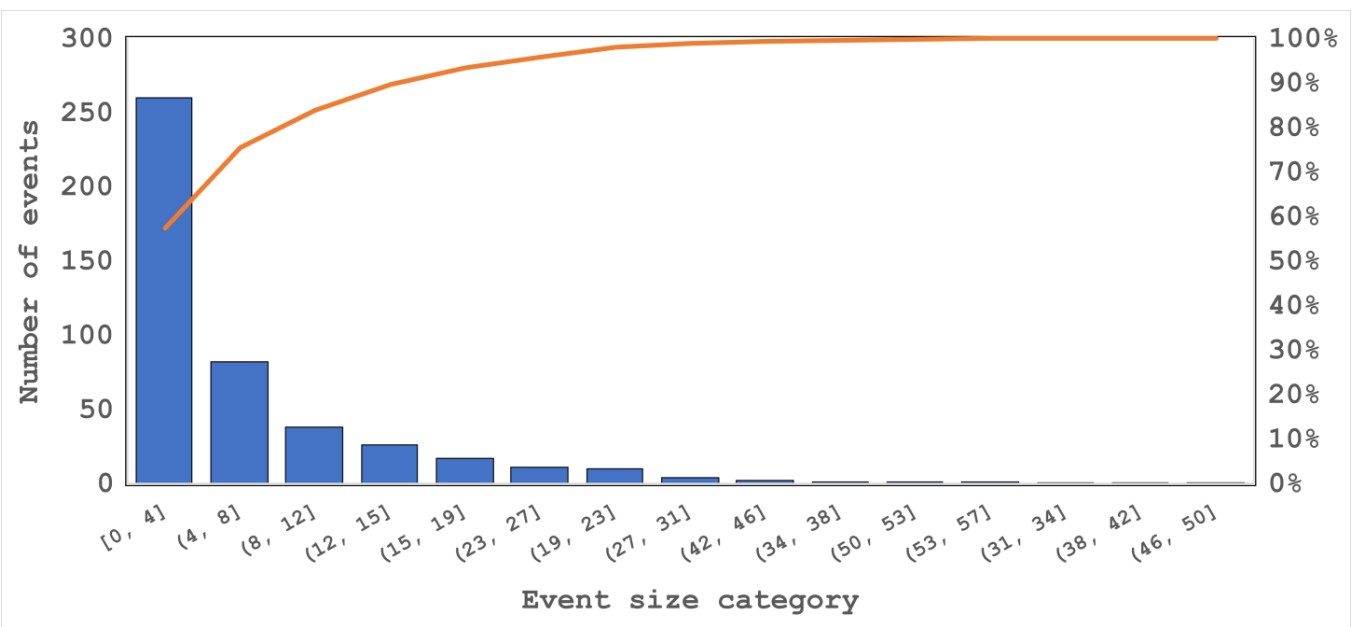

**Figure 7.** Pareto chart of single rain events from 2017 to 2023; event sum categories are represented in descending order by blue bars, and the cumulative total is represented by the red line.

There were no significant seasonal differences in mean rainfall amount per event ($p = 0.30$) with mean values of 6.3 mm in winter (DJF), 5.9 mm in spring (MAM), 6.4 mm in summer (JJA), and 4.4 mm in autumn (SON).

The mean runoff coefficients for single rain events $C_{se}$ during the whole measurement period of 2017–2023 were 0.20 for WRB85 and M60, 0.21 for Z100, and 0.22 for Z40 (no significant difference, $p = 0.81$). Mean peak reduction or peak discharge coefficients $C_p$ were 0.04 (M60), 0.03 (WRB85), and 0.02 (Z40 and Z100), and the mean detention times $T_d$ were 436 min (M60), 260 min (WRB85), 364 min (Z40), and 365 min (Z100).

The retention performance indicators of all BGR types were partly affected by the rainfall amount of each event and the season (Figure 8a–f). The mean values of runoff coefficients $C_{se}$ of all BGR types were significantly affected by the rainfall event size category (Figure 8a, $p < 0.05$) and season (Figure 8b, $p < 0.05$). For M60, the post-hoc test indicated significant differences in $C_{se}$ between event size categories < 4 mm and >12 mm ($p = 0.000$); for WRB85, between <4 mm and >12 mm ($p = 0.002$) and 4–12 mm and >12 mm ($p = 0.025$); for Z40, between <4 mm and >12 mm ($p = 0.000$) and 4–12 mm and >12 mm ($p = 0.000$); and for Z100, between <4 mm and >12 mm ($p = 0.000$) and 4–12 mm and >12 mm ($p = 0.000$). For M60, the post-hoc test also indicated significant differences in $C_{se}$ between all seasons ($p = 0.000$) except between spring and autumn ($p = 0.857$); for WRB85, the post-hoc test indicated significant differences in $C_{se}$ between all seasons ($p < 0.05$) except between winter and autumn ($p = 0.069$); for Z40, there were significant differences ($p < 0.05$) for all seasons except between spring and summer ($p = 0.384$) and spring and autumn ($p = 0.179$); and for Z100, there were significant differences ($p < 0.05$) for all seasons except between spring and summer ($p = 0.987$) and spring and autumn ($p = 0.168$).

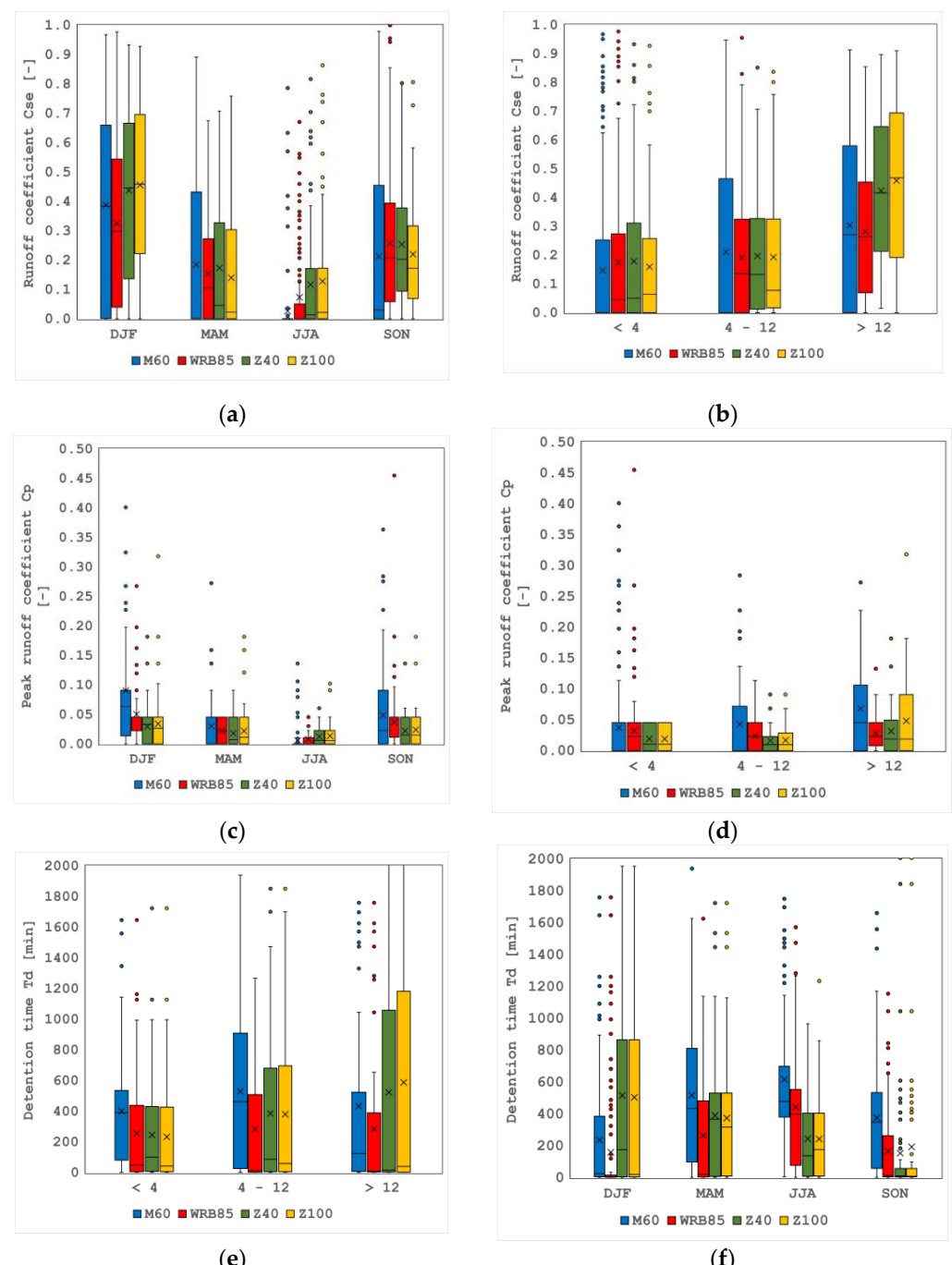

**Figure 8.** Box plots of mean runoff coefficients for single rain events $C_{se}$ (**a**) for different rain event size categories and (**b**) for winter (DJF), spring (MAM), summer (JJA), and autumn (SON) months, for peak runoff coefficients ($C_p$) in different event size categories (**c**) and seasons (**d**), and detention times ($T_d$) for event sizes (**e**) and seasons (**f**). Mean values are indicated by x, and horizontal lines in the boxes are medians.

Mean peak discharge coefficients $C_p$ differ significantly in rain event size categories for BGRs M60, Z40, and Z100 (Figure 8c, $p < 0.05$) and seasons for all BGR types (Figure 8d, $p < 0.05$). For M60, the post-hoc test indicated significant differences in $C_p$ between event size categories <4 mm and >12 mm ($p = 0.01$); for Z40, between <4 mm and >12 mm ($p = 0.003$) and 4–12 mm and > 12 mm ($p = 0.001$); and for Z100, between <4 mm and >12 mm ($p = 0.000$) and 4–12 mm and >12 mm ($p = 0.000$). For M60, the post-hoc test also indicated significant differences in $C_p$ between all seasons ($p = 0.000$) except between spring and

summer ($p$ = 0.063) and spring and autumn ($p$ = 0.254); for WRB85, the post-hoc test indicated significant differences in $C_p$ between all seasons ($p < 0.05$) except between winter and autumn ($p$ = 0.138), spring and summer ($p$ = 0.174) and spring and autumn ($p$ = 0.143); for Z40 there were significant differences ($p < 0.05$) for all seasons except between winter and autumn ($p$ = 0.207), spring and summer ($p$ = 0.451) and spring and autumn ($p$ = 0.682). For Z100, there were only significant differences ($p$ = 0.000) in $C_p$ between winter and summer.

The mean detention time is significantly different for the rain event size categories for BGR types M60, Z40, and Z100 (Figure 8e, $p < 0.05$) and in different seasons for all BGR types (Figure 8f, $p < 0.05$). For M60, the post-hoc test indicated significant differences in mean detention time between event size categories <4 mm and 4–12 mm ($p$ = 0.008); for Z40, between <4 mm and 4–12 mm ($p$ = 0.022) <4 mm and >12 mm ($p$ = 0.000) and 4–12 mm and >12 mm ($p$ = 0.045); and for Z100, between <4 mm and 4–12 mm ($p$ = 0.022) <4 mm and >12 mm ($p$ = 0.000) and 4–12 mm and >12 mm ($p$ = 0.007). For M60, the post-hoc test also indicated significant differences in mean detention time between all seasons ($p < 0.05$) except between winter and autumn ($p$ = 0.061) and spring and summer ($p$ = 0.282); for WRB85, the post-hoc test indicated significant differences in mean detention time between all seasons ($p < 0.05$) except between winter and spring ($p$ = 0.139), winter and autumn ($p$ = 1.000), and spring and autumn ($p$ = 0.121); for Z40, there were only significant differences ($p$ = 0.002) between winter and summer; the same was true for Z100 ($p$ = 0.005).

Therefore, in most cases, the hydrological performance (retention, peak discharge, and detention) of the BGRs was best for small rain events and in warm months with high potential evapotranspiration, which is also the case for conventional GRs [4].

Nevertheless, the BGRs were effective in preventing discharge from heavy rain events. The retention performance for heavy rain events can be described exemplarily by Figure 9. In this event from 27 August 2019, about 35 mm of rain fell within 1 h on the roofs. A rain event of this size typically occurs only once in 30 years in Hamburg. The extensive GR (SD) had the highest runoff ($C_{se}$ = 0.48) compared to the BR types with $C_{se}$ values of 0.31 (Z100), 0.27 (Z40), 0.15 (WRB85), and 0 (M60) after 1 day. The discharge from the extensive GR started earlier and with a high intensity (max. 1.4 mm/min), which decreased after 1 h. The throttle effect of the BGRs' outflow elements can be seen clearly from the low and nearly constant discharge intensity (<0.05 mm/min).

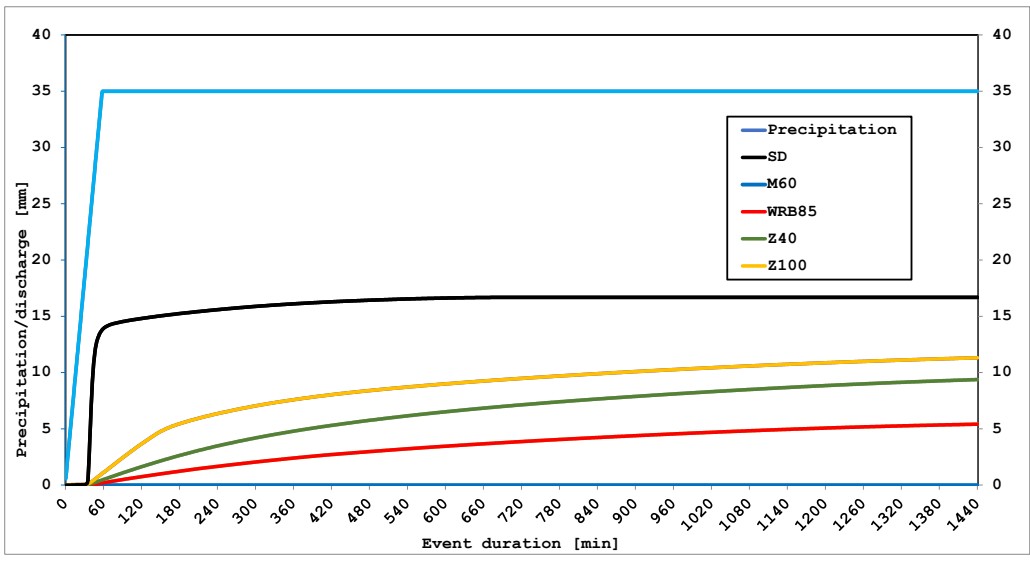

**Figure 9.** Cumulative curves of rain and discharge from different (B)GR types from the heavy rain event on 27 August 2019.

### 3.3. Vegetation Composition Analysis

After eight years of nearly undisturbed development, with maintenance occurring once to twice a year (removing tree sprouts, cleaning outlets and gravel verges), the roofs have developed differing appearances and biomass volumes (Figure 10a–c). The number of species on the different (B)GR types is used as an indicator of plant biodiversity in this study. In 2022, WRB85 (28 species), Z40 (25), and M60 (24) had comparable species biodiversity and significantly more than Z100 (16) and SD (14); see Table A1.

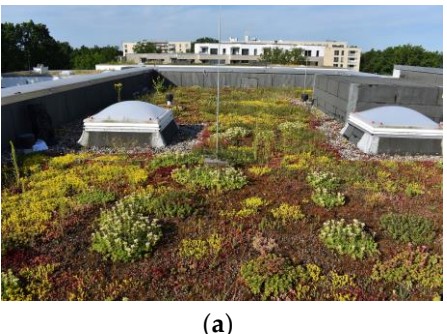 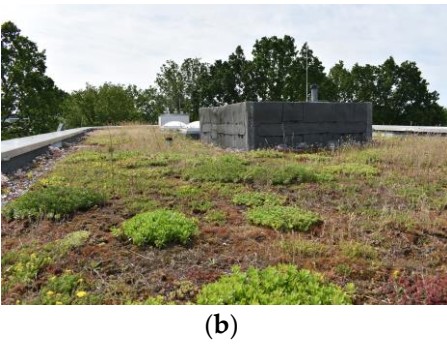 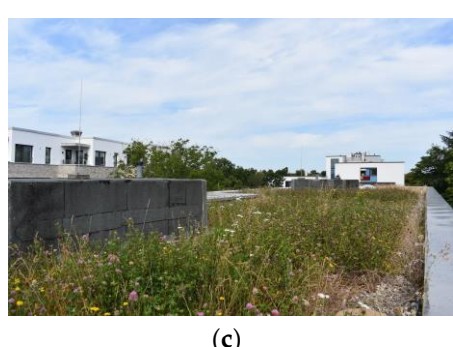

(**a**)          (**b**)          (**c**)

**Figure 10.** Photos of the different GR types at the research site in June 2022. Extensive green roof SD (**a**), blue-green roof Z40 (**b**), and blue-green roof WRB85 (**c**).

A total of 21 species, which were not expected in any of the seed mixtures of the roofs, have been detected and thus probably immigrated (species in Table A1). Of these 21 species, 13 were detected on WRB85, 10 on M60 and Z40, 9 on SD, and 6 on Z100. As can be seen in Table A1, there are species groups which can be assigned to different green roof types. Species at the top of the table in the green box (12 species) occur on most of the studied roof types. The next group (blue box) consists of 7 species only occurring on BGRs. The largest group of 15 species (grey) only grow on BGRs with long-term rainwater storage (M60 and WRB85) and could be benefiting from the better water supply and passive irrigation out of the storage layer. They would therefore possibly not be able to survive on the other roofs without long-term storage. The last species group (red) is only present on BGRs without long-term storage (Z40 and Z100).

The better water supply on the BGRs with long-term storage could be the reason for higher species diversity. The comparable high diversity on Z40, which has no long-term storage, could be due to the thicker substrate layer height (9 mm) than Z100 (6 mm) and therefore better water retention and a more diverse seed mixture ("Hamburg nature roof").

## 4. Discussion

### 4.1. Hydrological Performance of BGRs

To the knowledge of the authors, this is the longest scientific investigation of full-scale (static) BGRs up to now. The BGRs proved to be very effective in decreasing the runoff from the roofs and therefore increasing the evapotranspiration and approaching the natural water cycle. They can be one effective measure in mitigating urban flooding from heavy rain events. Retention performance indicators of runoff coefficients, peak runoff coefficients, and detention were improved considerably compared to conventional extensive roofs with comparable substrate thicknesses (6–9 mm). The long-term or yearly runoff coefficient $C_{lt}$ for extensive GRs can vary, e.g., from 0.12 [34]–0.88 [35], but is typically assumed to be 0.5 [19], which was confirmed by another long-term experiment on a full-scale extensive GR in Hamburg [36]. The studied BGRs had considerably lower $C_{lt}$ from 0.26–0.36, meaning that they had much more potential to increase evapotranspiration from roofs, even without long-term storage. In other studies, long-term runoff coefficients for BGRs of 0.35 (dynamic BGR, Palermo, Italy [26]), 0.25 (dynamic BGR, Viterbo, Italy [26]), 0 (Cagliari, Italy [26]), 0.18–0.19 (static BGR plot, Nunica, MI, USA [37]), and 0.51–0.37 (static BGR plots, Toronto,

ON, Canada [38]) were reported. The long-term runoff coefficients of the most of these BGRs and of this study correspond with those from intensive GRs with substrate heights of 250–500 mm, which is 0.3 [19]. The goal of a near-natural water balance or low long-term discharges could therefore be reached or even be overcompensated for by building BGRs (or intensive GRs). A better performance than conventional extensive GRs in reducing runoff coefficients for single rain events [39], along with peak runoff coefficients [21,39] and outflow rates [20], and delaying runoff for a longer time [38,39] are reported for several BGR types.

The higher hydrological efficiency of the BGRs is due to the 0° slope of the BGRs, which can lead to reduced runoff coefficients [5,13,23]. Furthermore, the very low peak runoff vales of the studied BGRs are due to the limited runoff intensities by the outlet elements. These elements can be decisive in bridging stress periods of the sewer system after heavy rain events and rapid runoff from non-greened sealed surfaces. All BGR types had these benefits, with or without the long-term storage of rainwater. In the case of long-term storage, of approximately 30 mm, the evapotranspiration capacity was increased compared to conventional green roofs or maintained even during dry periods [24,40]. Extensive green roofs can typically evaporate about 0.5–3.5 mm per day [41]. Evaporation can be increased on extensive GRs with additional (external) irrigation or infiltration or by using BGRs up to 5 mm per day [40,42]. If stormwater is discharged by "smart throttles" (dynamic BGRs) only immediately before potential heavy rainfall events, stormwater retention could be optimized, and the evapotranspiration performance of retention roofs can be maintained for even longer during dry periods [24]. In this way, the amount of retained (and evaporated) stormwater volume can approach up to 100% [25]. The more significant amount of stored rainwater and the buffered release help manage stormwater, reduce stress on sewer systems, and reduce flooding risk, independently of the specific BGR type (with/without storage, static/dynamic).

*4.2. Vegetation Composition Analysis*

The vegetation analysis indicates that the improved water supply can support species diversity. As mentioned earlier, increased water storage could replace thicker substrate layers for hydrological performance if desired. A key abiotic factor driving differences in species survivorship on green roofs is the growing medium's depth [43,44], due to reduced plant stress through increased water retention [45] and an increase in plant abundance, cover, and dry mass with an increase in the growing medium's depth [46]. BGRs were able to retain a higher level of substrate moisture for longer periods and [40] demonstrated that, except for a few hot and dry summer days during which the moisture content dropped below 10%, BGRs were able to maintain considerably higher soil moisture compared to other GR types. The vegetation structure and species composition also influence the stormwater retention of GRs. The higher the leaf area or leaf area index (LAI) of plants, the higher their interception capacity (temporary water storage on leaf surfaces). In addition, a more extensive root system increases water retention capacity and runoff delay due to an increased amount of water absorbed through the roots [47]. Accordingly, [47] rated the suitability for increasing the retention capacity of grasses the highest, followed by herbaceous species and sedum species. Species with higher evaporation rates lead to faster drying of the substrate and thus faster regeneration of the water uptake capacity [48]. For example, [49] recorded 60% higher daily evapotranspiration rates for grasses than for sedum on GRs. These observations argue for even greater hydrological efficiency through greater plant species diversity and more grass and herbs species on the studied BGRs.

Another performance-increasing factor could be the high organic matter content in the substrate due to several years of higher biomass turnover on the BGRs with long-term storage. High organic matter in substrates has been found to elevate evaporative cooling [50,51] and retention capacity [16]. The greater growth intensity and species changes on roofs with long-term storage could require added care and maintenance, which has not been the case on the studied roofs up to now.

*4.3. Future Research Directions*

The development of retention performance with increasing age of green roofs should be researched in greater depth to gather more information about aging processes and their effects on hydrological dynamics. This would also require studies on changes in the substrate, e.g., with respect to the accumulation of organic matter. Such aging dynamics still seem to be one of the greatest major unknowns in green roof research [22]. Especially for BGRs, a more precise description of the water balance should be sought. For static outflow elements, the dynamics of the water levels and a comparison to dynamic systems would be important to determine the necessity of further technologization and digitalization of the systems. Thus, conclusions could be drawn about the evaporation capacity and, accordingly, the prediction of emptying times. With regard to the provision of storage for extraordinarily heavy rains, there are still some uncertainties in this respect which also need to be clarified.

The evolution of species composition needs to be further monitored to understand the effect of increased biodiversity and the interaction with hydrological effects.

**5. Conclusions**

The studied BGRs can significantly reduce and delay runoff from roofs and increase their evapotranspiration. They are also effective in reducing flood risk from heavy rain events. The hydrological effectiveness and the urban climatic potential of BGRs for cooling the surrounding area appear to be higher due to the tendency of better water supply compared to conventional extensive green roofs. The increased water storage capacity allows more diverse vegetation to develop. If rainwater is permanently stored on the roof, the species composition can change in the long term, and the evaporation capacity and thus the cooling effect on the environment can be increased and maintained even during dry periods. For areas with a high building density, such BGRs have far-reaching positive effects beyond their hydrological aspects.

**Author Contributions:** Conceptualization, M.R. and W.D.; methodology, M.R.; formal analysis, M.R.; data curation, M.R.; writing—original draft preparation, M.R.; writing—review and editing, W.D.; visualization, M.R.; project administration, M.R.; funding acquisition, W.D. All authors have read and agreed to the published version of the manuscript.

**Funding:** This research was funded by the German Federal Ministry for the Environment, Nature Conservation and Nuclear Safety, grant number 03DAS032, and the Hamburg Authority for Environment and Energy.

**Data Availability Statement:** The data presented in this study are available on request from the corresponding author. The data are not publicly available due to the participation of the companies involved.

**Acknowledgments:** The authors would like to express their sincere thanks for the provision of the technical infrastructure, the roofs, and green roof materials to the SAGA corporate group, Zinco GmbH, and Optigrün International AG.

**Conflicts of Interest:** The authors declare no conflict of interest.

## Appendix A

**Table A1.** List of vegetation species composition and their relative abundance on different roof types. Numbers and letters in the species' rows indicate the Braun–Blanquet rating scheme for vegetation cover abundance (r: less than 1% plot cover, 3–5 individuals; +: <5%, few individuals; 1: <5%, numerous individuals; 2: 5–25%; 3: 25–50%; 4: 50–75%; 5: 75–100%). Endangered species are marked with * and underlined species have immigrated The green box indicates species occuring on most of the studied roof types, the blue box indicates species only occurring on BGRs, the grey box indicates species only occuring on BGRs with long-term rainwater storage (M60 and WRB85) and the red box indicates species only occurring on BGRs without long-term rainwater storage (Z40 and Z100).

| Roof Type | WRB85 | M60 | Z40 | Z100 | SD |
|---|---|---|---|---|---|
| Area [m$^2$] | 220 | 220 | 220 | 220 | 135 |
| Substrate height [mm] | 7 | 8 | 9 | 6 | 6 |
| Number of species | 28 | 24 | 25 | 16 | 14 |
| Sedum album | 2 | 2 | 2 | 2 | 3 |
| Sedum hispanicum | r | | 1 | 1 | 1 |
| Sedum sexangulare * | 2 | 2 | 1 | 1 | 2 |
| Sedum kamtschatikum | 2 | 2 | 2 | 1 | 2 |
| Sedum spurium | 2 | 2 | 2 | 1 | 2 |
| Taraxacum officinale | + | 1 | 1 | + | 1 |
| Trifolium dubium | | 1 | 1 | 1 | 1 |
| Trifolium pratense | 2 | 2 | r | | + |
| Holcus lanatus | | + | 1 | + | + |
| Hieracium pilosella * | + | r | r | + | r |
| Trifolium arvense | 1 | + | | | r |
| Silene vulgaris * | + | | + | + | |
| Dianthus deltoides * | 1 | 1 | + | r | |
| Potentilla argentea | 1 | r | + | | |
| Bromus hordeaceus | 1 | | 1 | 1 | |
| Epilobium spec. | + | r | | r | |
| Vicia parviflora | + | | r | | |
| Senecio inaequidens | + | | r | | |
| Achillea millefolium | + | 2 | | | |
| Leucanthemum vulgare | + | 1 | | | |
| Prunella grandiflora | r | + | | | |
| Prunella vulgaris | r | + | | | |
| Origanum vulgare | + | + | | | |
| Cirsium vulgare | r | r | | | |
| Plantago argentea | + | r | | | |
| Anthemis tinctoria | + | + | | | |
| Plantago lanceolata | + | | | | |
| Poa annua | r | | | | |
| Petrorhagia saxifraga | + | | | | |
| Veronica filiformis | r | | | | |
| Linaria vulgaris | | r | | | |
| Melilotus officinalis | | 1 | | | |
| Trifolium repens | | 2 | | | |

**Table A1.** *Cont.*

| Roof Type | WRB85 | M60 | Z40 | Z100 | SD |
|---|---|---|---|---|---|
| *Hypericum perforatum* | | | 1 | | |
| *Erodium cicutarium* | | | r | | |
| <u>*Stellaria media*</u> | | | r | | |
| *Saxifraga granulate* * | | | + | | |
| *Campanula spec.* | | | + | | |
| *Clinopodium vulgare* * | | | + | + | |
| *Festuca ovina* * | | | 1 | + | |
| *Crepis spec.* | | | + | + | 1 |
| *Senecio vulgaris* | | | r | | r |
| *Tragopogon pratensis* | | | | | r |

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
