# Peer review of "Long-Term Performance of Blue-Green Roof Systems—Results of a Building-Scale Monitoring Study in Hamburg, Germany"

_water, doi:10.3390/w15152806_

Round 1
Reviewer 1 Report
The topic chosen by the Authors for analysis seemed very interesting and relevant. To the knowledge not only of the authors and also reviewer, this is the longest scientific investigation of full-353 scale (static) BGRs until now. So far, research results on the hydrological performance of green roofs are mostly published after a few months of testing or even after a few rainfall events, without taking into account the change in their behaviour during operation. Here, the authors additionally include blue-green roofs, which are relatively new solutions for which research results are lacking. For these reasons, the content presented is of great value to both researchers and designers and future implementations.
The other minor comments are provided in the following:
1. It is unlikely to be the practice in scientific articles published in the journal Water to divide the Introduction chapter into sections. For these reasons, it is proposed to change the layout to one 'continuous' text.
2. Authors often use commercial names together with the company name, e.g. Optigrün BGRs , the systems "Flow Control Extensive" and "Meander FKM 60", "Optigrün Economy Roof", etc. I understand the thanks to the company in the Acknowledgments. However, in the reviewer's opinion, commercial names should not be used in scientific papers, as this could be seen as advertising for that company. I leave it to the editors to resolve this issue.
3. Page 14, line 433 is Tthe and should be The.
Apart from these comments, in the reviewer's opinion the article is maturely written, contains interesting research results and the authors have mastered the art of writing scientific articles.
Author Response
We would like to thank you for your very helpful and constructive comments. We have edited the comments as follows:
- The headings in the introduction have been removed
-
I can very well understand the concern for advertising and I can assure you that we tried to mention the companies and product names as rarely as possible. But we think it is better for the readers if they know exactly which products were studied. Therefore, we have kept the product and company names. It should be clear that in our study we are neutral towards the companies and act scientifically independent, which we also declare by excluding any conflict of interest. We also mention the companies and products only in the "methods" chapter and have deliberately used only the abbreviations (Z40, Z100, M60, WRB85) in the other parts to not mention the companies too often. And in the discussion and conclusion, the companies and the specific products are not named, but the effects of the different BGR types are generalized for similar types (static BGRs with/without long-term water storage).
- Page 14, line 433, Tthe has been changed into The
Reviewer 2 Report
This in an interesting study dealing with the performance of the blue and green roof building monitoring systems. I have few suggestions about the work and these suggestion may be very helpful to the authors. These are as follows:
1. The abstract section needs to be revised. The abstract should address objectives, methods, results and policy implications properly. I missed it.
2. The introduction is very nicely arranged. I would suggest to incorporate few literatures in the introduction sections.
https://doi.org/10.1016/j.trd.2023.103723.
https://doi.org/10.1016/j.iref.2023.06.033.
3. There could be more different types of the building materials? Building type should be presented in Table for better understanding.
4. Statistical analysis should be described properly. PostHOC tukey test could be also performed.
5. Table2 should be placed in the supplementary section.
NA
Author Response
Thank you for the helpful comments on our manuscript. We have incorporated these as follows:
- We added several aspects in the abstract (lines 9 - 21) and have made it clearer what our objectives, methods, results and possible policy implication were.
- In our opinion, the suggested literature "Zhengtao Qin, Yuan Liang, Chao Yang, Qingyan Fu, Yuan Chao, Ziang Liu, Quan Yuan (2023) Externalities from restrictions: Examining the short-run effects of urban core-focused driving restriction policies on air quality, Transportation Research Part D: Transport and Environment,Volume 119." and "Binbin Yu, Xinru Zhou (2023)
Urban administrative hierarchy and urban land use efficiency: Evidence from Chinese cities, International Review of Economics & Finance,
Volume 88" do not really fit into the thematic focus of our study: Therefore, we have not considered them to incorporate into our introduction. - We added the sentence "The buildings were 3 storey apartment blocks made of reinforced concrete construction. " (line 151) to clarify the building type.
- We described the statistical analysis in more detail (lines 251 - 257) and performed post-hoc Tukey tests to find out which groups are significantly different from each other (lines 293 - 294 and 326 - 364).
- We placed the table in the appendix.
Reviewer 3 Report
Thank you for the opportunity to review such interesting, useful, and actual paper. Research on rainwater management in urban areas is obviously necessary. Sustainable exploitation of water resources and their protection is of key importance for smart development. The manuscript presents reviewing the state of the art of green roofs and hydrological evaluation of different type of green roofs located in Hamburg. The paper falls within the scope of Water journal. The outcomes of the research can be beneficial for designing strategies of rainwater management.
• Abstract & Introduction: These two parts are focused on the paper's main aim and the new contributions of authors to the state of the art. The abstract with keywords very effectively summarizes the manuscript.
• Materials & Methods: Based on the very good basement of choosen research plan, the authors showed the topic from different points of view.
• Results & Discussion: The data are well-presented with relevant and current tables, figures, and references.
The article can be accepted in its current form, after taking into account one comment: The article needs an editorial correction, for example:
It should be written 74% not 74 % (Line 11)
The dot at the end of the sentence is missing (Line 122)
5. Conclusions (Line 435) - should be marked as main point of the manuscript
Author Response
Thank you very much for your comments. We made several editorial corrections, we added blanks in front of every % and every unit. We added the missing dot at the end of line 122. And we marked the Conclusions heading (Line 435) as main point of the manuscript.
Round 2
Reviewer 2 Report
The manuscript has been revised as suggested. Now, it has gained the quality.